# Transgressive Behaviours and Readiness to Engage in Interreligious Dialogue in Young Adulthood

**Elżbieta Rydz [1,\*] and Daria Stawarz [2]**

1   Faculty of Social Sciences, Institute of Psychology, The John Paul II Catholic University of Lublin, 20-950 Lublin, Poland
2   Lublin Psychodynamic Center—Psychotherapy, 20-017 Lublin, Poland
\*   Correspondence: elzbieta.rydz@kul.pl

**Abstract:** One of the key aspects of an individual's spiritual intelligence is the ability to transcend, which allows one to identify dimensions of reality that go beyond the boundaries of the material world. In the present study, we look at transcendence from the theoretical perspective of a conception proposed by Kozielecki, who defines human transgressive behaviour as all those actions and acts of thinking which exceed the limits of an individual's existing material, symbolic and social capacities, and achievements and which constitute a source of new important values. One important social value is the readiness to engage in dialogue with people of other faiths. We wanted to investigate the associations between transgression and young people's readiness to enter into such dialogue. Rydz and Bartczuk, departing from a psycholinguistic definition of dialogue, developed a definition of readiness to engage in interreligious dialogue, describing it as a person's mental readiness to exchange views about religious topics with people of other religions. We hypothesised that there was a relationship between transgressive behaviours and dimensions of readiness to engage in interreligious dialogue among young adults. To verify these hypotheses, 528 people aged 18–25 were surveyed using the *Readiness to Engage in Interreligious Dialogue Test* (REID) and *the Alternative Behaviours Checklist* (ABC) based on Kozielecki's conception of transgression. The results we obtained show that the dimensions of social transgression, creative transgression, and psychological transgression were positively related to dimensions of readiness to engage in interreligious dialogue.

**Keywords:** spiritual intelligence; dimension of transgression; readiness to engage in interreligious dialogue; young adulthood

## 1. Introduction

One of the key aspects of spiritual intelligence is the ability to transcend (King 2008), which allows one to identify dimensions of reality, i.e., ones that cross the boundaries of the material world, as one comes to discern a reality that binds, permeates, and gives a meaning to various dimensions of actual existence (King 2008; cf. Emmons 2000). From transcendent awareness that goes beyond simple physicality comes another sense that provides a necessary point of reference for the actions one takes and the events one experiences.

In his transgressive conception of the human being, Kozielecki (2001) defines an individual's transgressive behaviour as the ability to transcend oneself (personal transcendence), which allows one to perform actions and acts of thinking that exceed the limits of one's existing capacities and material, symbolic and social achievements, i.e., a transcendence that is the source of new important values. In today's society, which is becoming more and more multicultural and multi-religious, one such socially important value is the readiness to engage in interreligious dialogue (Rydz et al. 2020).

## 2. Transgression

Theorists of spiritual transcendence/transgression believe that this ability is a key aspect of spiritual intelligence. Without going into too much detail about the contemporary

concepts and models of spirituality, we can say that the most famous conceptions of spiritual intelligence include the theoretical proposals by Emmons (2000), Zohar and Marshall (2001) and King (2008).

Emmons (2000) has identified five characteristics of a spiritually intelligent person, namely: (1) the capacity for transcendence as the ability to elevate one's consciousness above one's own physicality and perceive other dimensions of existence, (2) the ability to enter into heightened spiritual states of consciousness, (3) the ability to invest everyday activities, events, and relationships with a sense of the sacred and to reinforce motivation through a sense of a higher goal, (4) the ability to use spiritual resources to solve problems, e.g., by discerning a higher sense in them or finding consolation by seeing them from a religious perspective. Emmons observes that these skills are accompanied by adequate conduct, such as engaging in virtuous behaviours—forgiveness, gratitude, modesty, or compassion.

Zohar and Marshall (2001) define spiritual intelligence as having the following nine characteristics: (1) flexibility as the ability to actively and spontaneously self-adapt, (2) a high degree of self-awareness, (3) a capacity to face and use suffering, (4) a capacity to withstand and transcend pain, (5) sensitivity to inspiring visions and values, (6) a reluctance to cause unnecessary harm, (7) a tendency to see the connections between diverse things, (8) a strong need to ask *why?* questions, and (9) field independence.

King (2008) distinguishes four key aspects of spiritual intelligence:

(1) Critical existential thinking, described as the ability to critically contemplate the nature of existence, reality, universe, space, time, death, and other existential or metaphysical issues; (2) personal meaning production, defined as the ability to produce personal meaning and life purpose with regard to all physical and mental experiences; finding meaning in life increases one's motivation to act, confidence about the direction of one's development, and a sense of stability and order; it gives one a general sense of the purposefulness of one's existence; (3) transcendental awareness, understood as having the capacity to identify the transcendent dimensions of the Self, other people, and the physical world during normal states of consciousness, while being able to discern these dimensions and their relationship to the self and material reality; this ability involves perceiving a deeper reality that binds and permeates the individual layers of the material world and connects them in a structure of interdependencies, as well as discerning the relation of this reality to the Self; (4) conscious state expansion refers to the intentional capacity to enter and exit higher/spiritual states of consciousness at one's own discretion; higher states of consciousness are entered into by expanding one's field of attention, e.g., through meditation, which may lead to a better understanding or an increased empathy.

When analysing the above models, one notices that transcendent awareness/the capacity for transcendence is described as a key feature of spiritual intelligence, as the basic mechanism that allows one to go beyond one's current physical, mental, and social condition, and adds to those domains of one's life a deeper dimension by launching the process of identification of transcendent dimensions of reality and fusing them with the current life experience.

The concepts of transcendence and transgression have been developed in numerous psychological trends: the psychodynamic trend (Jung 1997; Fromm 1996), the humanistic and existential trend (Maslow 1990; Rogers 2002; Frankl 1984; Dąbrowski 1979), and the cognitive trend (cf. Atkinson 1957; Elliot 1999; Kozielecki 2001).

Kozielecki (2001), in his cognitive conception of psychotransgressionism, defined transgression as "all actions and acts of thinking—usually intentional and conscious—that cross the boundaries of a person's existing material, symbolic and social capacities and achievements, and become a source of important new positive and negative values", (Kozielecki 2001, p. 18) as well as "the process of thinking and engaging in practical action aimed at crossing the boundaries of the space and time in which a person has been active up till the present moment" (Kozielecki 1997, p. 59). Kozielecki emphasised that transgression concerned intentional, deliberate actions of the type "I know that I can".

With regard to the content of an individual's goals, transgression can manifest itself on the symbolic, psychological, social, and material plains (Kozielecki 1987, 2001): (1) symbolic transgression ("towards symbols"), also known as intellectual transgression, involves expanding one's knowledge about the world and creating new, often ingenious mental structures; it plays an important role in religion, philosophy, and art; (2) psychological transgression ("towards the self") is aimed at self-development, self-creation; it involves conscious work on one's willpower or traits of character; (3) social transgression ("towards others") may, on the one hand, be directed at one's altruistic and pro-social activities, and, on the other hand, at extending control and domination over other people; (4) material transgression ("towards things") is directed towards the physical; it manifests in expanding one's territory or in producing new goods.

Moreover, Kozielecki also distinguishes transgressive behaviours manifested in one's internal and personal development, which he calls personal transgression. Sometimes (especially in the case of creative transgression), the crossing of individual boundaries is consequential to a greater number of people and has a historical significance. Such behaviours fall under the category of historical transgression. If transgressive behaviours are displayed by a larger group of people, we are dealing with social transgression. In the case of small groups, we talk about collective transgression, in the case of groups consisting of thousands or millions of people—mass transgression. Expansive transgression consists of expanding the area over which one holds control, one's physical territory, and managing one's own weaknesses, while creative transgression involves discovering what has so far been unknown (inventions, concepts). Kozielecki also discerns emancipatory transgression, which he defines as expanding the scope of one's freedom (both subjective and objective), and temporal transgression, i.e., prolonging or perpetuating life by using diets, discovering medications, or creating works of art that keep individuals alive symbolically in other people's memory. When positive or negative effects of transgression are adopted as a criterion, transgression can be divided into constructive and destructive. The reversibility of the change that has occurred as a result of transgression can be used as another criterion of division. This creates the categories of reversible and irreversible transgression. Spontaneous changes, the emergence of new properties within a given system, based on a group process, is a specific transgression called emergence. There are also paradoxical transgressions, which involve a reduction in possessions or values. One can give up things one holds important for the sake of a higher goal, e.g., faith, another person's life, or personal change. Of course, one transgressive behaviour may belong to several categories (Kozielecki 1987, 1997, 2001, 2004; Ślaski 2012).

According to Kozielecki, all these types of change are primarily motivated by the hubristic need to defend, reassert, and increase one's own value as a person (Kozielecki 2004). This need is a super-need that regulates intentional development.

Kozielecki contrasts transgression with protective behaviours. The latter are conservative, adaptive behaviours, which are defined as everyday and routine activities that allow one to keep one's body in physical or psychological balance, e.g., maintaining health, ensuring personal or material security, etc. (Kozielecki 2001). By engaging in protective behaviours, one satisfies one's needs and gains a sense of security. Such activities are common and schematic and are performed within certain limits. They are often repeated cyclically and become habits. They reduce the tension caused by a lack or a potential lack and are used to maintain homeostasis. Since people find such activities indispensable, they can be categorised as "I know that I must" actions; Kozielecki describes them as normal and adaptive behaviours (Kozielecki 2004).

## 3. Readiness to Engage in Interreligious Dialogue

Representatives of the psychology of dialogue (cf. Buber 1954; Rogers 2002) claim that, to establish a dialogue, one needs to take a subjective approach to another person without treating them instrumentally and looking at them through the prism of one's own goals. Dialogue also requires going beyond social roles and adapting to the cultural context. An

attitude like this involves self-awareness, non-judgemental openness, and the ability to show psychological intimacy. Authenticity, empathic understanding, and unconditional acceptance are also some of the conditions that promote dialogue. Feller and Ryan (2012) have identified eight characteristics of dialogue in the context of social cohesion and cultural integrity: coexistence, movement, encountering "the other", assumption, creativity and flexibility, sharing, holistic, and multigenerational. According to those authors (2012, p. 357), dialogue is a movement whose goal is to generate coexistence through encountering the "other", to share experiences, to think together in creative and flexible ways, and to explore assumptions together. Dialogue is holistic; it requires sustained effort to engage a broad base of society, spanning all generations. Theme-oriented interpersonal dialogue is an example of social dialogue. Gorsky and Caspi (2005) have defined this type of dialogue as "a discursive relationship between two participants characterised by thought-provoking activities such as hypothesising, questioning, interpreting, explaining, evaluating, and rethinking issues or problems at hand" (p. 140). These definitions postulate the acceptance of the dignity and freedom of every human being and their right to express their views (even if they are objectively wrong) on account of ideological diversity, freedom of conscience, or religion (Łukaszyk 1979).

Dialogue can play an important role in searching for areas of and prospects for human progress, while strengthening the attitude of respect for all cultural values, including religious ones. As a consequence, it can increase the sense of security, reduce individual and group anxiety, and help avoid and solve conflicts (cf. Byrka et al. 2016). Research findings confirm that experiences of intercultural social interactions play an important role in shaping openness to interactions with the religious "other" (Burch-Brown and Baker 2016). Byrka et al. (2016), who analysed the domain of dialogue in democratic society using Clark's (1996) linguistic framework, described dialogue as a more or less symmetrical act of communication between two parties engaged together in the process of establishing a mutual understanding of what is being said.

This definition served as a starting point for the development of an original definition of the construct of "readiness to engage in interreligious dialogue" by Rydz and Bartczuk (Rydz et al. 2020). Departing from the psycholinguistic definition of dialogue (Clark 1996), the authors of the project developed a definition of the construct of readiness to engage in interreligious dialogue, which refers to one's mental readiness to start a conversation on religious topics with a person of another faith. This definition covers four aspects of dialogue: (1) dyadicity, which presupposes a certain level of general interest in religion, (2) symmetry, which requires an attitude of tolerance and respect for others and acceptance of one's own and other people's right to have and express personal views, (3) understanding, which implies interest in others and the willingness to get to know and understand them, and (4) commitment, which is based on a positive motivation to cooperate with people of other faiths and reflects the readiness to jointly pursue goals and take actions to reach an agreement (Rydz et al. 2020).

Research shows that people naturally make a better impression on members of their own group than on members of other groups, and that they favour individuals with whom they share a common cultural or group identity (Brewer 1979; Efferson et al. 2008). From the perspective of social identity theory, individuals prefer people with the same identity characteristics as their own, for instance, they show more favourable attitudes or allocate more resources to members of their in-group than to members of an out-group (Rubin and Hewstone 1998).

Constructs whose meaning is close to that of readiness to engage in interreligious dialogue include religious tolerance (e.g., Ekici and Yucel 2015; Hook et al. 2017; Putnam and Campbell 2010; Van der Straten Waillet and Roskam 2013; Van Tongeren et al. 2016), interreligious favourability (e.g., Ciftci et al. 2015), ecumenism as a dimension of a seeking attitude (Beck and Jessup 2004), attitude towards religious diversity (e.g., Francis et al. 2012; Gawali and Khattar 2016), xenosophia (Streib and Klein 2014, 2018; Streib et al. 2010), and prejudice in interreligious relations (Eka Putra 2016; Hunsberger and Jackson 2005;

Klein et al. 2018). Most of them, however, concern a global attitude towards the followers of other religions (the religious other).

Previous studies show that an important starting point for dialogue between particular social groups is the interlocutors' interest in each other and the culture of other groups, their history, customs, and religion. Research shows that openness to religious diversity is related to cognitive openness, need for cognition (Watson et al. 2015), open-mindedness (Gawali and Khattar 2016), and intellectual humility (Hook et al. 2017). Religious tolerance is positively correlated with openness to experience, imagination, artistic sensitivity, rich emotionality, cognitive curiosity, flexibility of behaviour, non-dogmatism, and internal religiosity (Kruglanski and Webster 1996; Altemeyer and Hunsberger 1992; Van der Straten Waillet and Roskam 2013; Deslandes and Anderson 2019; Van Tongeren et al. 2016). The attitude towards religious diversity was shown to correlate positively with open-mindedness and flexibility (Gawali and Khattar 2016). Religious openness, religious tolerance, and religious pluralism were negatively associated with cognitive closure, dogmatism, authoritarianism, submission to authority, general aggressiveness, and adherence to social conventions (Kruglanski and Webster 1996; Altemeyer and Hunsberger 1992; Van der Straten Waillet and Roskam 2013; Deslandes and Anderson 2019; Van Tongeren et al. 2016). Tolerance towards religiously dissimilar groups is negatively correlated with security-focused religious beliefs and positively correlated with growth-focused religious beliefs (quest religiousness) (Van Tongeren et al. 2016). Saroglou (2002) suggests that spiritual–emotional religiosity tends to be associated with a low need for closure, whereas religious fundamentalism and classical religiosity are associated with the need for closure.

In the literature on the subject, there is a clear gap in research on the relationship of religious tolerance and the readiness to engage in interreligious dialogue with spirituality and spiritual intelligence.

A study on interfaith spirituality revealed that this variable was moderately positively associated with a general religiosity measure (regardless of religious orientation) and with reappraisal, which plays a role in regulating emotions and reinterpreting events (Kira et al. 2021).

In research on prejudice and meta-stereotypes (i.e., in-group members' beliefs about how they are perceived by an out-group) that included two groups of participants, believers and non-believers (Saroglou et al. 2011), a number of tendencies regarding meta-stereotypes on personality traits were detected in both studied groups. Respondents from both groups tended to exaggerate their meta-stereotypes and to deny the out-group's main characteristics.

Research on the determinants and functions of spiritual intelligence revealed that among young participants of a course about consciousness, those who had a high level of spiritual intelligence were more open to different concepts of self-awareness and showed more capacity for self-reflection (Green and Noble 2010; Albursan et al. 2016). Khoshtinat (2012) noted that students' spiritual intelligence correlated positively with their religious adjustment. The results of a study by Munawar and colleagues (Munawar and Omama 2018) showed a significant correlation between spiritual intelligence, religiosity, and life satisfaction among elderly Pakistani Muslims, both women and men.

An investigation on existential quest (EQ), defined as "being open to questioning and changing one's own existential beliefs and worldviews" (Van Pachterbeke et al. 2012), revealed that EQ was negatively correlated with dogmatism and need for closure. Participants with high EQ scores were less prone to display myside bias both in terms of the number of arguments they generated and the level of conviction with which they supported them.

Summing up, it can be observed that in areas in which cultures mix and cultural boundaries are crossed, there emerges in society a new quality of awareness and a new quality of interactions between cultures, and sometimes even new values are created (cf. Sadowski 1999). The ability to elevate one's consciousness above one's own physicality and perceive other dimensions of existence, referred to as transcendence or transgression, is one of the key dimensions of emotional intelligence (Emmons 2000; King 2008). According to

the assumptions of Kozielecki's conception (Kozielecki 1987, 2001), the ability to transgress allows one to intentionally and consciously cross the boundaries of one's own material, symbolic and social capacities, and achievements, which become a source of new important values (Kozielecki 2001). One socially important value is the readiness to engage in interreligious dialogue with followers of different religions. This paper explores the relationships between transgressive behaviours and the readiness to engage in interreligious dialogue.

## 4. Method

### 4.1. Goal

The aim of this study was to identify the associations between transgressive behaviours and readiness to engage in interreligious dialogue.

### 4.2. Research Questions

The research problem was formulated in the form of two questions:

1. Is there a relationship between transgressive behaviours and the dimensions of readiness to engage in interreligious dialogue?
2. Do transgressive behaviours predict the dimensions of readiness to engage in interreligious dialogue?

### 4.3. Research Hypotheses

The following hypotheses were formulated on the basis of the literature of the subject.

**H1.** *There is a relationship between transgressive behaviours and the dimensions of readiness to engage in interreligious dialogue.*

**H2.** *Transgressive behaviours are predictors of the dimensions of readiness to engage in interreligious dialogue.*

### 4.4. Instruments

Two measures were used:

#### 4.4.1. The Readiness to Engage in Interreligious Dialogue Test (REID), by Rydz and Bartczuk, (Rydz et al. 2020)

REID by Rydz and Bartczuk was created on the basis of the authors' concept of interreligious dialogue. The test consists of 36 items rated on a six-point scale: −3—definitely not, −2—no, −1—probably not, 1—probably, 2—yes, 3—definitely. The items make up four subscales: (1) Readiness to Exchange Views on Religious Topics, (2) Readiness to Seek Mutual Understanding, (3) Readiness to Communicate with Followers of Other Religions, and (4) Barriers to the Symmetry of Dialogue.

REID1 Readiness to Exchange Views on Religious Topics includes (a) starting conversations about religious topics, (b) having a subjective sense of being open in a religious conversation, (c) having a subjective sense of finding it easy/difficult to share thoughts about faith.

REID2 Readiness to Seek Mutual Understanding covers (a) openness to understanding another person's religious views, (b) respect for views other than one's own, (c) ability to listen to another person's position to the end, (d) belief that good communication (conversation) can help resolve ambiguities and religious conflicts.

REID3 Readiness to Communicate with Followers of Other Religions includes (a) finding it easy to interact socially with people who hold different religious views than one's own, (b) interest in conversing with people holding different religious views, (c) finding it easy to take the perspective of a person with different religious views,

REID4 Barriers to the Symmetry of Dialogue covers (a) having difficulty accepting views different from one's own, (b) feeling discomfort when confronted with a person with different religious views, (c) feeling anger when confronted with a person having different

religious views, (d) having a sense of superiority when confronted with a person having different religious views.

The tool is theoretically valid. Cronbach's alpha reliability coefficients for the individual scales are as follows: REID1 $\alpha = 0.91$, REID2 $\alpha = 0.89$, REID3 $\alpha = 0.80$, and REID4 $\alpha = 0.82$ (Rydz et al. 2020). Reliability was also measured for the present sample (N = 528); the following reliability indicators were obtained: REID1 $\alpha = 0.86$, REID2 $\alpha = 0.89$, REID3 A$\alpha = 0.82$, and REID4 $\alpha = 0.80$.

4.4.2. Alternative Behaviours Checklist (ABC), by Ślaski (2010)

The Alternative Behaviours Checklist (ABC, Lista zachowań alternatywnych, in Polish) is a questionnaire developed by Ślaski (2010) on the basis of Kozielecki's conception according to which a human being is a telic system capable of transgressive behaviours that consist in going beyond one's own limitations and achieving in various spheres of life (Kozielecki 1987; Ślaski 2010).

A factor analysis with Oblimin rotation revealed eight factors which explained 47.2% of the total variance (Ślaski 2010). They included

(1) creative transgression—conscious exceeding of one's own supra-personal and social achievements and innovatively approaching heretofore unsolved problems; it is related to creativity in various areas of human activity;
(2) psychological transgression—conscious exceeding of one's personal capacities in the psychological sphere; it is associated with psychological self-improvement;
(3) social transgression—conscious exceeding of one's personal achievements in the social and public sphere; it is aimed at changing the reality in this area;
(4) ethical transgression—conscious exceeding of one's personal achievements in the moral sphere; it is associated with ethical improvement;
(5) material transgression—conscious exceeding of one's personal capacities in the material sphere, constantly enlarging one's financial and material resources;
(6) occupational transgression—conscious exceeding of one's personal capacities in the occupational sphere, acquiring new work-related skills;
(7) family transgression—conscious exceeding of one's personal capacities in the family sphere (life partner, spouse, children), acquiring new skills allowing an individual to coexist in the family;
(8) protective behaviours, which allow one to maintain somatic and mental balance and protect one's material and professional resources; they are aimed at maintaining the current state of affairs (Ślaski 2012).

The questionnaire contains 61 items rated on a four-point scale, where: 0—not at all accurate, 1—somewhat accurate, 2—moderately accurate, 3—very accurate, 4—completely accurate.

The instrument has satisfactory validity and reliability. The Cronbach's alpha reliability coefficient for the transgression dimensions ranged from 0.72 to 0.86. The reliability was measured for the purpose of this study in a group of subjects (N = 528), and the reliability coefficient ranged from 0.73 to 0.90, except for the modified subscale of family transgression ($\alpha = 0.29$) and protective behaviour ($\alpha = 0.47$). Cronbach's alpha coefficients for the group of participants we surveyed were calculated using a version of the questionnaire that had been modified to adjust the content of the test items to the developmental stage of the age group under study (18–25 years): minor changes were made to the items related to family transgression. The response option "not applicable" was added to five test items (items no. 7, 17, 23, 50, 59). In addition, seven job-related test items (items no. 9, 22, 26, 30, 33, 52, 58) were supplemented so that they also referred to school and academic studies.

With the consent of the authors of the ABC, we prepared an online version of the checklist using a Google sheet.

## 5. Participants

A total of 528 people aged between 18 and 25 were surveyed, including 302 women (57.2%) and 226 men (42.8%). The mean age was 21.5 years, and the standard deviation was 2.04. The respondents were in their early adulthood. They came from all of Poland's 16 provinces (voivodeships). Four participants (0.8%) were Poles living outside their homeland (the Netherlands, England, Iceland, and Scotland). Most of the respondents lived in the following provinces: Lubelskie (31.8%), Podkarpackie (30.7%), Małopolskie (9.7%), Mazowieckie (8.9%), and Dolnośląskie (4.9%). The most numerous group of participants (36.7%) came from large cities with populations of over 250,000 inhabitants; 29.9% of the respondents lived in villages, 18.9% in towns below 50,000 inhabitants, and 14.2% in medium-sized towns of 50,000–250,000 inhabitants.

As far as marital status was concerned, 52.7% of the respondents declared they were single, 40.5% lived in an informal relationship, 4.2% were married, and 2.7% lived in a religious congregation or seminary.

The participants' main occupations were university studies (64.6%), work (18.9%), secondary and tertiary-level schools, including: 6.8% high schools, 2.7% technical colleges, and the remaining 7% included a vocational school, an art school, a post-secondary school, and others.

Our sample included people enrolled in various educational programmes. A total of 22.7% studied social sciences, 21.8% science, 17% liberal arts, 11.6% medical sciences, and 8.1% artistic sciences. A total of 5.9% went to a vocational school, 1.9% were in a philosophy programme, and 11% declared they followed a program other than the ones listed above.

The respondents were asked what religion they identified with (and not what religion they were formally affiliated with). The responses revealed the following three most numerous groups: Roman Catholics (83.7%), atheists (6.8%), and agnostics (4.7%). The remaining 4.8% of the surveyed identified with other religious groups. They included pagans (neo-pagans, native Slavic believers, and followers of shamanism) (0.8%), Buddhists (0.6%), Greek Catholics (0.6%), Eastern Orthodox Christians (0.4%), Unaffiliated Christians (0.4%), Protestants (0.2%), Jews (0.2%), and Jehovah's Witnesses (0.2%). Individuals who described themselves as not identifying with a specific religion or declared they professed other religions (belief in the existence of a Higher Being, belief in inner energy, an informal home Church, a religion combining the ideas of Christianity, Islam, and Buddhism) accounted for 0.8% of the sample.

For the item asking the participants how strong their faith was on a scale from 0 to 10, the mean score was 6.87, and the standard deviation was 2.82. The participants' percentage religious affiliation was analogous to the distribution reported in the analyses of this phenomenon conducted by the Central Statistical Office in Poland for this age group. The Central Statistical Office's data also show that this age group has the smallest percentage of deeply religious people (4.8%) and the largest percentage of people who declare themselves as non-believers (approx. 5%). A relatively large percentage of people in this age range are seeking and undecided (12.5%) or indifferent to religion (7.3%). The remaining part (70.02%), who are believers, is comparable in size to other age groups (Bieńkuńska and Ciecieląg 2018).

Many more respondents declared that they had contacted followers of other religions in person (87.5%) than via the Internet (31.1%).

As many as 45% of the participants had been abroad five or more times. Participants who visited another country once or twice represented 23.7% of the sample. Slightly fewer—22% of the respondents—declared they had been abroad three or four times, and those who had never left their homeland accounted for 8.7% of the sample. The most frequent responses to the question about the length of the stay abroad were: less than 2 months (32.8%), less than 6 months (25.2%), 6 months and more (19.9%), less than 7 days (13.8%), and not at all (8.3%).

At the end of the survey, the respondents were asked to rate their wellbeing on a scale from 0 to 9. The mean score was 6.3, and the standard deviation was 1.85.

## 6. Procedure

The survey was conducted online using a Google sheet. A group of 528 respondents who met the age criteria were included in the study. The respondents were recruited by snowball sampling. The survey was conducted before the COVID-19 pandemic.

## 7. Results

Data were analysed statistically using IBM SPSS Statistics 25 software. The results were considered significant at $\alpha = 0.05$.

### 7.1. Correlation Analysis

In order to identify the relationships of the eight types of transgression and protective behaviours with the four REID subscales, we ran a Pearson's correlation (r) test (Table 1).

**Table 1.** Pearson's correlations (r) between Transgression (TR) and Readiness to Engage in Interreligious Dialogue (REID) (N = 528).

| | REID1 Readiness to Exchange Views on Religious Topics | REID2 Readiness to Seek Mutual Understanding | REID3 Readiness to Communicate with Followers of Other Religions | REID4 Internal Barriers to the Symmetry of Dialogue |
|---|---|---|---|---|
| TR_C1 Creative Transgression | 0.296 ** | 0.217 ** | 0.290 ** | −0.108 * |
| TR_P2 Psychological Transgression | 0.282 ** | 0.228 ** | 0.246 ** | −0.174 ** |
| TR_E3 Ethical Transgression | 0.361 ** | 0.215 ** | 0.152 ** | 0.059 |
| TR_S4 Social Transgression | 0.359 ** | 0.341 ** | 0.287 ** | −0.140 ** |
| TR_M5 Material Transgression | 0.055 | −0.048 | 0.077 | 0.034 |
| TR_F6 Family Transgression | 0.112 * | 0.150 ** | 0.041 | 0.001 |
| TR_O7 Occupational Transgression | 0.171 ** | 0.096 * | 0.184 ** | −0.084 |
| TR_PB8 Protective Behaviours | −0.054 | −0.081 | −0.176 ** | 0.166 ** |

* correlation significant at the level of 0.05 two-tailed; ** correlation significant at the level of 0.01 two-tailed.

The REID dimension of the Social Transgression scale was statistically significantly, weakly positively correlated with REID1—Readiness to Exchange Views on Religious Topics ($r = 0.359$, $p < 0.01$), REID2—Readiness to Seek Mutual Understanding ($r = 0.341$, $p < 0.01$), and REID3—Readiness to Communicate with Followers of Other Religions ($r = 0.287$, $p < 0.01$), and showed a very weak negative correlation with REID4—Internal barriers to the Symmetry of Dialogue ($r = −0.140$, $p < 0.01$).

Similarly, the dimension of Psychological Transgression was weakly, but statistically significantly, correlated with all the REID scales: positive correlations with REID1—Readiness to Exchange Views on Religious Topics ($r = 0.282$, $p < 0.01$), REID2—Readiness to Seek Mutual Understanding ($r = 0.228$, $p < 0.01$), REID3—Readiness to Communicate with Followers of Other Religions ($r = 0.246$, $p < 0.01$), and a weak negative correlation with REID4—Internal Barriers to the Symmetry of Dialogue ($r = −0.174$, $p < 0.01$).

Creative Transgression was also weakly positively correlated with all the Readiness scales: REID1 ($r = 0.296$, $p < 0.01$), REID2 ($r = 0.217$, $p < 0.01$), REID3 ($r = 0.290$, $p < 0.01$), and weakly negatively correlated with the Barriers scale REID4 ($r = −0.108$, $p < 0.05$).

For the dimension of Ethical Transgression, significant weak positive correlations were obtained with REID1 ($r = 0.361$, $p < 0.01$), REID2 ($r = 0.215$, $p < 0.01$), and REID3 ($r = 0.152$, $p < 0.01$). The ethical dimension of transgression did not correlate with REID4.

For the Family Transgression dimension, significant weak positive correlations were obtained with the REID1 ($r = 0.112$, $p < 0.05$) and REID2 ($r = 0.150$, $p < 0.01$).

Moreover, some correlation trends were observed for Occupational Transgression and Protective b/Behaviours.

Occupational Transgression correlated significantly, but very weakly, with the three Readiness scales: REID1—Readiness to Exchange Views on Religious Topics (r = 0.171, $p < 0.01$), REID2—Readiness to Seek Mutual Understanding (r = 0.096, $p < 0.01$), and REID3—Readiness to Communicate with Followers of Other Religions (r = 0.084, $p < 0.01$).

Protective Behaviours correlated very weakly with two REID scales: negatively with REID3 (r = −0. 176, $p < 0.01$) and positively with REID4 (r = 0.166, $p < 0.01$). The directions of the relationships for Protective Behaviours were opposite to the directions of the relationships obtained for the Transgression scales.

*7.2. Regression Analysis*

To predict the individual REID dimensions on the basis of the eight dimensions of transgressive behaviours and protective behaviours, a stepwise regression analysis was performed.

Five variables entered the model in which the dependent variable was readiness to exchange views on religious topics: ethical, social, creative, and occupational transgression, and protective behaviours. This model turned out to be well-fitted to the data F (5.522) = 24.552; $p < 0.01$. The predictive power of ethical transgression was beta = 0.235; $p < 0.01$, social transgression beta = 0.201; $p < 0.01$, creative transgression beta = 0.185; $p < 0.05$, occupational transgression beta = −0.152; $p < 0.05$, and protective behaviours beta = −0.087; $p < 0.05$. These values of the coefficients can be interpreted as showing that the higher the severity of ethical, social, and creative transgression, the higher was the participant's readiness to exchange views on religious topics. By contrast, the higher the level of occupational transgression and protective behaviours, the lower was the young people's readiness to talk about religious topics. This model explained 18.3% of the variance (Table 2).

**Table 2.** Regression analysis of the Transgression dimensions which explain Readiness to Exchange Views on Religious Topics (N = 528).

| Readiness to Exchange Views on Religious Topics | | | | |
|---|---|---|---|---|
| | **Beta** | **T** | **F** | **R2** |
| Ethical Transgression | 0.235 ** | 4.766 | 78.862 | 0.129 |
| Social Transgression | 0.201 ** | 3.799 | 51.822 | 0.162 |
| Creative Transgression | 0.185 * | 3.190 | 28.452 | 0.172 |
| Occupational Transgression | −0.152 * | −2.744 | 24.552 | 0.183 |
| Protective Behaviours | −0.087 * | −2.183 | 36.426 | 0.168 |

R2—adjusted R-squared. All F values are statistically significant. Results for predictors that significantly explained the level of the dependent variable are shown; the variables that did not enter the model were psychological transgression, material transgression, and family transgression. *—significant at $p < 0.05$; **—significant at $p < 0.01$.

Three variables entered the model in which the dependent variable was readiness to seek mutual understanding: social, material, and creative transgression. This model turned out to be well-fitted to the data F (3,524) = 29.113; $p < 0.01$. It should be mentioned that social transgression was a stronger predictor (beta = 0.311; $p < 0.01$) than material transgression (beta = −0. 181 $p < 0.01$) and creative transgression (beta = 0.131; $p < 0.05$) (Table 3). The coefficients we obtained show that the higher the level of social and creative transgression, the greater the participants' readiness to seek mutual understanding. Material transgression was the only variable to correlate negatively with the dependent variable: the higher the participants scored on material transgression, the less willing they were to seek mutual understanding. This model explained 13.8% of the variance (Table 3).

**Table 3.** Regression analysis of the Transgression dimensions which explain Readiness to Seek Mutual Understanding (N = 528).

| | Readiness to Seek Mutual Understanding | | | |
|---|---|---|---|---|
| | **Beta** | **T** | **F** | **R2** |
| Social Transgression | 0.311 ** | 6.432 | 69.006 | 0.114 |
| Material Transgression | −0.181 ** | −3.954 | 40.251 | 0.130 |
| Creative Transgression | 0.131 * | 2.462 | 29.113 | 0.138 |

R2—adjusted R-squared. All F values are statistically significant. Results for predictors that significantly explained the level of the dependent variable are shown; the variables that did not enter the model were psychological transgression, ethical transgression, occupational transgression, protective behaviours, and family transgression. *—significant at $p < 0.05$; **—significant at $p < 0.01$.

Four variables entered the model in which the dependent variable was readiness to communicate with followers of other religions: social transgression, creative transgression, and protective behaviours. This model turned out to be well-fitted to the data $F (4,523) = 22.804$; $p < 0.01$. The strongest predictor was social transgression (beta = 0.226; $p < 0.01$), followed by: creative transgression (beta = 0.188 $p < 0.01$) and protective behaviours (beta = −166; $p < 0.01$) (Table 4). The coefficients demonstrate that the higher the participants' level of social and creative transgression, the more willing they were to communicate with the followers of other religions. It was also predicted that the higher the level of protective behaviours, the lower would be the participants' readiness to communicate with believers in other religions. This model explained 13.7% of the variance (Table 4).

**Table 4.** Regression analysis of the Transgression dimensions which explain Readiness to Communicate with Followers of Other Religions (N = 528).

| | Readiness to Communicate with Followers of Other Religions | | | |
|---|---|---|---|---|
| | **Beta** | **T** | **F** | **R2** |
| Creative Transgression | 0.188 ** | 3.835 | 48.287 | 0.082 |
| Protective Behaviours | −0.166 ** | −3.964 | 33.275 | 0.109 |
| Social Transgression | 0.226 ** | 4.573 | 28.887 | 0.137 |

R2—adjusted R-squared. All F values were statistically significant. Results for predictors that significantly explained the level of the dependent variable are shown; variables that did not enter the model were psychological transgression, ethical transgression, material transgression, and occupational transgression. **—significant at $p < 0.01$.

The last model, in which the dependent variable was internal barriers to the symmetry of dialogue, included five variables: psychological, ethical, social, and material transgression and protective behaviours. This model turned out to be well-fitted to the data $F (5,522) = 15.296$; $p < 0.01$. The predictive power of the independent variables was beta = −0.308 $p < 0.01$ for psychological transgression, beta = 0.307; $p < 0.01$ for ethical transgression, beta = −0.181; $p < 0.05$ for social transgression, beta = 0.156; $p < 0.05$ for protective behaviours, and beta = 0.136; $p < 0.05$ for material transgression. The coefficients show that the higher the level of material transgression, ethical transgression, and protective behaviours, the greater are internal barriers to the symmetry of dialogue. By contrast, the higher the level of psychological and social transgression, the lower are internal barriers to the symmetry of dialogue. The model explained 11.9% of the variance (Table 5).

**Table 5.** Regression analysis of the transgression dimensions which explain Iinternal barriers to the Ssymmetry of dialogues (N = 528).

| | Internal barriers to the Symmetry of Dialogue | | |
| --- | --- | --- | --- | --- |
| | **Beta** | **T** | **F** | **R2** |
| Psychological Transgression | −0.308 ** | −5.378 | 16.399 | 0.028 |
| Ethical Transgression | 0.307 ** | 5.705 | 18.971 | 0.064 |
| Protective Behaviours | 0.156 * | 3.789 | 18.372 | 0.090 |
| Social Transgression | −0.181 * | −3.372 | 16.607 | 0.106 |
| Material Transgression | 0.136 * | 3.006 | 15.296 | 0.119 |

R2—adjusted R-squared. All F values were statistically significant. Results for predictors that significantly explained the level of the dependent variable are shown; variables that did not enter the model were creative transgression, occupational transgression, family transgression. *—significant at $p < 0.05$; **—significant at $p < 0.01$.

## 8. Discussion

An important aspect of spiritual intelligence is the ability to transcend (King 2008), which allows one to identify transcendent dimensions of reality that go beyond the boundaries of the material world by discerning a deeper reality that binds, permeates, and gives a deeper meaning to various dimensions of the real world.

According to Kozielecki's theory of psychotransgressionism (Kozielecki 1987, 2001), the ability to transgress allows one to intentionally and consciously cross the boundaries of one's own material, symbolic, and social capacities and achievements, which become a source of new important values. Transgression can manifest itself as transgressive behaviours at the symbolic, psychological, social, or material level (Kozielecki 2001). An important value in multicultural societies is the readiness of individuals to enter into interreligious dialogue, that is, their mental preparedness to start a conversation on religious topics with a person of another faith, a conversation that is dyadic and symmetrical and whose participants show mutual understanding and involvement. (Rydz et al. 2020).

Two research questions were asked: Is the capacity for transgression related to an individual's readiness to engage in interreligious dialogue? Which transgressive behaviours predict the development of readiness to engage in interreligious dialogue?

The goal of this study was to analyse the relationships between transgressive behaviours and the dimensions of readiness to engage in interreligious dialogue.

The first hypothesis was partially confirmed.

Weak associations were obtained between the dimension of social transgression and all the REID dimensions: readiness to exchange views on religious topics, readiness to seek mutual understanding, readiness to communicate with followers of other religions, and internal barriers to the symmetry of dialogue (a very weak negative correlation). This means that people who are more capable of consciously surpassing their own achievements at a personal level in the social and public sphere and are more inclined to make effort toward changing their social reality (social transgression) tend to be more willing to communicate on religious topics, which manifests in their readiness to start conversations on those topics, their subjective sense of personal openness in a conversation on religious topics, and a subjective feeling that it is easy for them to share their reflections on faith. They may also be inclined to seek mutual understanding in an exchange with a person of another religion, i.e., try to be open to understanding the religious views of their interlocutor, respect their different perspective, hear out their position, and believe that good communication can help resolve religious ambiguities and conflicts. Moreover, people with higher social transgression scores may be more inclined to communicate with representatives of other religions, i.e., they may find it easier to interact with people having different religious views, show interest in conversing with them, or adopt their perspective.

Our results also demonstrate that people who are high on social transgression may show a tendency towards lower barriers to the symmetry of dialogue.

　　　　　　　　　　　　　　　　　　　　　　　　　　　　　

These findings are consistent with Kozielecki's conception (Kozielecki 1987, 2001, 2004), in which social transgression is described as the ability to reach out to others in pro-social activities in many spheres of social life, including the religious sphere. The ability to enter into dialogue with people who hold different views is highly valued socially as it fosters good relations between groups and the exchange of ideological values, enriching people's own vision of the world/universe, helping them develop awareness of different perspectives from which to define the same reality, maintain social ties, create a social identity, and avoid misunderstandings or explain their causes (e.g., Byrka et al. 2016; Hamilton and Wills-Toker 2006; Kozlovic 2003; Mutz 2006). Our results suggest that a higher level and spiritual intelligence in the dimension of social transgression may favour the development of a readiness to engage in interreligious dialogue. In an analogous study of existential quest (EQ), a construct defined as "being open to questioning and changing one's own existential beliefs and worldviews" (Van Pachterbeke et al. 2012), EQ was found to correlate negatively with dogmatism and need for closure, and positively with lower susceptibility to myside bias.

Similarly, statistically significant weak correlations were obtained for the dimension of psychological transgression with all the REID dimensions: positive for readiness to exchange views on religious topics, readiness to seek mutual understanding, and readiness to communicate with followers of other religions, and negative for internal barriers to the symmetry of dialogue.

In his transgressive conception of the human being, Kozielecki (1987) emphasised that transgressive behaviours are intentional, deliberate actions, and that psychological transgression is geared towards self-development and self-creation. One can consciously work on one's willpower and character traits. According to Kozielecki, the readiness to change is primarily motivated by the hubristic need to defend, reassert, and increase one's own value as a person, which regulates an individual's development (Kozielecki 2004). Previous studies on the determinants and functions of spiritual intelligence revealed that young people with a high level of spiritual intelligence were more open to concepts of self-awareness different from their own and more capable of self-reflection (Albursan et al. 2016), while in adults, spiritual intelligence played the role of a buffer in existential crises (Skrzypińska and Drzeżdżon 2020). Results of other studies on the relationship between an individual's traits and attitudes towards followers of other religions show that cognitive openness, cognitive need (Watson et al. 2015), mental openness (Gawali and Khattar 2016), and intellectual humility (Hook et al. 2017) promote an attitude of openness to religious diversity. Similarly, traits such as openness to experience, imagination, artistic sensitivity, rich emotionality, cognitive curiosity, and flexibility of behaviour correlate positively with religious tolerance (Kruglanski and Webster 1996; Altemeyer and Hunsberger 1992; Van der Straten Waillet and Roskam 2013; Deslandes and Anderson 2019; Van Tongeren et al. 2016). These conclusions are congruent with the tendencies we found in the relationships between psychological transgression and readiness to engage in an interreligious dialogue with people of other faiths. These associations are analogous to those between social transgression and interreligious dialogue. From the point of view of the psychology of purposive behaviour, socially important values may be the source and prototype of one's goals which develop in the process of internalisation (Nuttin 1984; Nurmi 1994). The high social value of social dialogue, including interreligious dialogue, may be a basis for intentional transgressive behaviours (psychological transgression). This speculation requires further theoretical and empirical explorations.

Similar tendencies were observed for the dimension of creative transgression, which, according to Kozielecki (1987, 2001), is conscious transgression of one's own achievements on an impersonal and social level, involves an innovative approach to previously unsolved problems, and is related to creativity in various spheres of human activity.

In our study, creative transgression was weakly positively correlated with all the Readiness scales of the REID test and very weakly negatively correlated with internal barriers to the symmetry of dialogue.

According to Kozielecki's transgressive conception of the human being (Kozielecki 1996, 2001), for individuals with a transgressive orientation, transgression is a prerequisite for the expansion and inventive creation of the world. Kozielecki notes that spiritual transgression is associated with an individual's freedom and points out that freedom is a potential transgression, and a transgression is freedom realised (Kozielecki 1996). Creative transgression is related to solving problems in various fields of knowledge, especially artistic and philosophical ones. It involves looking for various pieces of information and points of view in order to generate several possible solutions to a problem. Thus, the association of this variable with the readiness to talk about religious topics can be explained by the fact that religious dialogue provides an opportunity to seek knowledge in order to solve problems, including theological and philosophical ones.

A similar position is taken by Buzan (2001), who defines spiritual intelligence as a spiritual state characterised by creativity, which makes a person become joyful, tolerant, and persistent. The correlation trends between creative transgression and readiness to engage in interreligious dialogue we observed in this study suggest that creative transgression can be related to higher tolerance, in this case, religious tolerance expressed in a dialogue with persons of another faith.

The dimension of ethical transgression, defined as the conscious exceeding of one's personal achievements in the moral sphere to improve one's ethical thinking and behaviour, was found to be significantly, though weakly, positively correlated with readiness to exchange views on religious topics and very weakly (also positively) with the readiness to seek mutual understanding and readiness to communicate with followers of other religions. The ethical dimension of transgression did not correlate with barriers to the symmetry of dialogue.

The trends we observed were consistent with our expectations based on Kozielecki's theory of transgression (Kozielecki 1987). Emmons (2000) takes a similar position in his conception of spiritual intelligence. He describes a spiritually intelligent person as having the capacity to invest daily activities, events, and relationships with a sense of the sacred, reinforce motivation through a sense of a higher goal and the capacity to use spiritual resources in problem solving, for example, by discerning a higher sense in them or finding consolation by seeing them from a religious perspective. Emmons observes that these skills are accompanied by adequate conduct, such as engaging in virtuous behaviours—forgiveness, gratitude, modesty, or compassion. Similarly, in their conception of spiritual intelligence, Zohar and Marshall (2001) distinguish ethical aspects of an individual's behaviour, such as sensitivity to inspiring values and opposition to unnecessary suffering.

Buzan (2001) also suggests that spiritual intelligence is a natural extension of personal and social intelligence, self-awareness, and appreciation of others. In Van Pachterbeke, Keller, and Saroglou's study (Van Pachterbeke et al. 2012) cited earlier in this paper, participants with high EQ scores were less prone to display myside bias both in terms of the number of arguments they generated and the level of conviction with which they supported them.

The statistically significant positive correlations were obtained for family transgression with two REID dimensions: readiness to exchange views on religious topics and readiness to seek mutual understanding. This means that people who scored high on family transgression, understood as deliberately exceeding one's own capacities at the personal level in the family sphere and acquiring new skills allowing an individual to coexist in the family, were more willing to exchange their religious views with others, which was manifest in their readiness to start conversations on religious topics, their subjective sense of personal openness in a conversation on religious topics, and a subjective feeling that it was easy for them to share their thoughts about faith. These findings are new and interesting in that they show that people who have higher family transgression capacities also have better communication skills when it comes to religious topics, with the reservation that this dimension of dialogue concerns openness to exchanging religious views with others, but not necessarily with people of other faiths. It can be assumed that the family

environment is a place of daily practice of dialogue in various spheres of life. In Poland, religion can be one of the topics of everyday dialogue, as evidenced by the correlations we obtained between the dimension of family transgression and the dimension of readiness to exchange views on religious topics, which involves starting conversations on religious topics, having a subjective sense of being open to such exchanges, and finding it easy to share one's thoughts about faith. Taking into account the young age of the respondents and their marital status (only 4.2% of them were married), most of them must have used their family of origin and not their own relationship as a point of reference in responding to the survey questions.

Occupational transgression correlated significantly but very weakly with three REID dimensions: readiness to exchange views on religious topics, readiness to seek mutual understanding, and readiness to communicate with followers of other religions. These results should be viewed as correlation trends. In future explorations, it is worth taking a closer look at these trends, as the social environment outside the family—at university or at work—provides numerous opportunities for interacting with people who hold worldviews and beliefs that are different from one's own, which, according to social dialogue research, increases one's ability to engage in dialogue with people of different views. Previous research demonstrates that experiences of intercultural social interaction, including in the work environment, play an important role in shaping one's openness to interactions with people holding different religious views (Burch-Brown and Baker 2016; Lando et al. 2015).

Protective behaviours correlated very weakly with two REID scales: negatively with readiness to communicate with followers of other religions, and positively with internal barriers to the symmetry of dialogue. These results can be treated as correlation trends. The directions of the relationships for protective behaviours are opposite to the directions of the relationships obtained for the transgression scales.

A hypothesis was also put forward regarding the predictive function of transgressive behaviours in relation to the REID dimensions. The hypothesis was partially confirmed.

The model, in which readiness to exchange views on religious topics was the dependent variable, included ethical transgression, social transgression, creative transgression, occupational transgression, and protective behaviours. The correlation coefficients we obtained showed that the higher the level of ethical, social, and creative transgression, the higher was the readiness to exchange views on religious topics. By contrast, the higher the level of occupational transgression and protective behaviours, the lower was the readiness to talk about religious topics. This model explained 18.3% of the variance.

The model which explained readiness to seek mutual understanding included social transgression, material transgression, and creative transgression. The coefficients we obtained suggest that the higher the level of social and creative transgression, the greater one's readiness to seek mutual understanding. Material transgression was the only type of transgression to correlate negatively with the dependent variable: the higher the level of material transgression, the lower was one's readiness to seek mutual understanding. This model explained 13.8% of the variance.

The model in which the dependent variable was readiness to communicate with followers of other religions contained four independent variables: social transgression, creative transgression, and protective behaviours. An interpretation of the coefficients led to the conclusion that the higher the level of one's social and creative transgression, the higher was one's readiness to communicate with believers in other religions. It was also predicted that individuals with high levels of family transgression and protective behaviours would be less ready to communicate with followers of other religions. This model explained 13.7% of the variance.

The last model, which explained internal barriers to the symmetry of dialogue, included psychological transgression, ethical transgression, social transgression, material transgression, and protective behaviours. When interpreting the coefficients, it can be seen that the higher the level of ethical and material transgression and protective behaviours, the greater were the internal barriers to the symmetry of dialogue. By contrast, high levels

of psychological and social transgression were predictive of lower internal barriers to the symmetry of dialogue. This model explained 11.9% of the variance.

Summing up, it can be observed that two transgression dimensions in particular—social transgression and creative transgression—were good predictors of readiness to engage in interreligious dialogue. An analysis of the role of the dimensions of transgression as predictors of internal barriers to the symmetry of dialogue shows that social transgression and psychological transgression lower internal barriers to the symmetry of dialogue, while protective behaviours, ethical transgression, and material transgression raise those barriers. Family transgression promotes openness to exchanging views on religious topics (positive correlations), preferably with people with similar religious beliefs, and not necessarily with people of another religion (family transgression along with protective behaviours were negative predictors of readiness to communicate with followers of other religions).

In a similar study conducted by Ślaski (2012), relationships were sought among early adults between transgressive behaviours and defensive self-consciousness, which is a trait opposite to readiness to enter into dialogue with a person holding a different worldview. That study revealed, among others, that creative, material, and occupational transgression were negatively correlated with defensive self-consciousness, while protective behaviours were significantly positively correlated with that trait (Ślaski 2012).

Our research touches upon key new aspects of individuals' transgressive behaviours in the context of the social challenges posed by the need for interreligious dialogue.

Research on personal, ethnic, and religious tolerance carried out in 2009–2016 among Polish students shows that personal contact with and expanding knowledge about "other" people increases tolerance (Szczęch and Rostek 2016). Analysing these results from the perspective of cognitive psychology, one can explain this phenomenon in terms of anticipatory schemas. The human mind uses the knowledge it has acquired to create a cognitive schema that allows one to predict, with greater or lesser probability, things such as someone's behaviour. The more knowledge one has and the more complete one's schemas are, the fewer emotions, including fear, are evoked by a stimulus (cf. Falkowski 2001; Bandura et al. 2001). Studnicki (2016) emphasises the role of imagination, which helps one to empathise with one's interlocutor's psychological needs.

On the one hand, the readiness to enter into interreligious dialogue is a manifestation of a broader predisposition to transgress. On the other hand, transgressive behaviours are governed by the law of growth (Kozielecki 2001; Ślaski 2012). This means that one's first successful attempts at transgressive behaviour, e.g., entering into interreligious dialogue, are likely to become an incentive for further development, and thus for adopting a more and more transgressive attitude in life. According to the author of the cognitive conception of psychotransgression (Kozielecki 2001), one's level of openness to transgression affects one's readiness to undertake transgressive actions, and the more one engages in transgressive behaviours, the more transgressive one's personality becomes.

The study reported in this paper has some limitations. First of all, it was a correlation study, a type of research design that allows one to analyse relationships but not to make cause-and-effect inferences. Future investigations should therefore be supplemented with experimental or qualitative studies. Moreover, our results should be interpreted with great caution as the correlation coefficients we obtained are weak. They do, however, show clear trends in the investigated relationships, providing inspiration for further research in this area.

Future studies should also take into account some additional personal and demographic variables, such as long-term, daily interactions with people of different faith.

The present study was conducted in a sample of young Poles; in future explorations, the research population should include participants at all stages of adulthood representing other nationalities and cultures. We assume that the models of relationships may be different in different age groups, and age may prove to be an important moderator of the associations between transgression and readiness to engage in interreligious dialogue, given the intensive developmental processes that occur at young ages (such as consolidation

of identity, including religious identity) and the accumulation of life experiences and the distinct nature of these experiences in middle and late adulthood.

**Author Contributions:** Conceptualization, E.R. and D.S.; methodology, E.R. and D.S.; software, D.S.; validation, E.R. and D.S.; formal analysis, E.R. and D.S; investigation, D.S.; resources, E.R. and D.S.; data curation, D.S.; writing—original draft preparation, E.R. and D.S.; writing—review and editing, E.R.; visualization, D.S.; supervision, E.R.; project administration, E.R. and D.S. All authors have read and agreed to the published version of the manuscript.

**Funding:** This research received no external funding.

**Institutional Review Board Statement:** The study was conducted in accordance with the Declaration of Helsinki and approved by Ethics Committee for Scientific Research, Institute Psychology of The John Paul II Catholic University of Lublin, protocol code: KEBN_25/2020, date of approval: 22 June 2020.

**Informed Consent Statement:** Informed consent was obtained from all subjects involved in the study.

**Data Availability Statement:** Data supporting reported results are available in the Institutional Repository of the John Paul II Catholic University of Lublin at the link http://hdl.handle.net/20.500 .12153/3906 (accessed on 10 November 2022).

**Conflicts of Interest:** The authors declare no conflict of interest.

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
