# Peer review of "Transgressive Behaviours and Readiness to Engage in Interreligious Dialogue in Young Adulthood"

_religions, doi:10.3390/rel13121194_

Round 1

Reviewer 1 Report

Review

“Transgressive behaviours and readiness to engage in interreligious dialogue in young adulthood”  

In general.

It was a pleasure to read the manuscript. The manuscript is well written, clear in structure and well-grounded in theory. The title covers the content. The topic is interesting and suits the journal. The introduction contains an extensive review of theories on spiritual intellect and transgressive behaviour and should be shortened (limited to a theory suiting the empirical investigation: suggestions follow). The research questions are clear and answered by an empirical investigation. Hypotheses are not well-formulated/according to the state of art and should be revised. The method is sound and the conclusions are supported by the findings. The correlations found although statistically significant are low and require some more relativizing remarks. The manuscript offers an empirical investigation by a questionnaire into the capability of individuals to engage in an interreligious dialogue which is relevant for science and contemporary society. Therefore, my advice is to publish the paper in its current form while paying attention to the following questions and suggestions. Shortening the theory in the introduction and revising the hypotheses (page 7) would improve the quality of the manuscript.

I have the following questions and suggestions:

Tables 2-5: There are no * and ** in the tables, while the descriptions indicate statistically significance. Please put asterisks in these tables where of application.  

p1. Introduction: “…One of the key aspects of spiritual intelligence is the ability to transcend (King 2008), which allows one to identify transcendent dimensions of reality…” This definition is circular. Please do not use the definiendum (transcend) in the definiens (transcendent dimensions) and what is the meaning of “deeper”? Although the authors cite King here, I think from a logical point of view the definition is not adequate and should be modified by the authors.    

p1 “…. a deepened sense”. What is meant by deepened? 

p2, paragraph 6: “…adds to those domains of one's life a deeper dimension…” Again, the word deeper raises questions. Maybe, replace it by “another”?

p3. paragraph 2-4. Do we really need these paragraphs in order to come to the research question? Consider to delete them or explain in one sentence why we need them.  

p4. paragraph 2-3. Idem. Consider deleting them. 

p4. I think the paragraph on dialogue can be shortened. The paper does not need a review on dialogue but a definition serving the empirical research. Consider to start this section with paragraph 2 on page 5.  

p4, paragraph 3: “…and seek social justice and peace…”. I wonder of this is always the case. Some seek power or war by dialogue. Do the authors have a specific kind of dialogue in mind?

p7. What is the difference between research question 1 and 2? If there is a significant relationship between two variables, one of them can be considered a predictor of the other. In the method section, a correlation study and regression analysis are proposed as study method. Here (p7), the meaning of a correlation between scales or the prediction by a regression model should be specified.

p7. The hypotheses are repeating the research questions, while they should be deduced from theory and translated into specific measurements. Please reframe the hypotheses. Make them express relationships between aspects of transgressive behavior and dimensions of readiness to engage in interreligious dialogue, for example less barriers higher readiness to engage,  etc. 

p8: “The questionnaire contains 61 items rated on a four-point scale, where: 0 – not at all accurate, 1 – somewhat accurate, 2 – moderately accurate, 3 – very accurate, 4 – completely accurate.” How was the score on a subscale calculated? Sum of item score or some cutoff per item and sum of items?

p7. Please explain why the measurements can be considered to represent continuous and not discrete variables.   

p8. Could the characteristics of the participants be captured by a table?

p9. Participants. What was the non-response and how could it have influenced the sample?

p10. “The largest number of relationships …” Please explain. Other scales are also correlated with all the REID dimensions. It is not the number of correlations that counts, but the association as such.

p11, paragraph6: “These results should be viewed as correlation trends.” Please explain. Also, for p15, paragraph 4. 

p11, paragraph 7: “These results can be treated as correlation trends.” Please explain. “ The directions of the relationships for Protective Behaviours were opposite to the directions of the relationships obtained for the Transgression scales.” I think the pattern is as expected. Is that what the authors mean by trend?

p15, paragraph 4. The results are as expected, but what does statistically significant low correlations as such mean? In general, there must be some other factor connected with the dependent variable. What could such a factor in case of transgressive behavior be? The nature of the adherent (catholic) faith?

p16: “The ethical dimension of transgression did not correlate with barriers to the symmetry of dialogue.” How can this finding be explained? A negative correlation could be expected. 

p18: “Moreover, our results should be interpreted with great caution as the correlation coefficients we obtained are weak.” This points out other stronger factors not art of this study might be correlated with the dependent variable. Could the author point out what such a factor can be? 

p18. “They do, however, show clear trends in the investigated relationships…” Again, I think the word trend is not appropriate. I read it as trend in time, while I think the authors mean pattern. 

p19: “age may prove to be an important moderator of the associations…” Might age be a factor explaining the low correlations found?

My advice: accept the paper while paying attention to the remarks being made.  

Author Response

Response to Reviewer 1 comments  (attached).

Reviewer 2 Report

The article is clear and soud methodologically, its well written, concise. The theories and methods are in line with the maximalist requirements of science.

The applied measures to survey transgressions. The utilized tool is "The Readiness to Engage in Interreligious Dialogue Test REID", Rydz & Bartczuk,

(Rydz et al., 2020), which is applied perfectly. The article is based on original research, conducted vis surveying on the internet. The analysis of the results is pristine, the presentation is above the average. I think this is a high level article, well written, scientifically justified. The journal is compatible with this high level of literary introduction, individual research and scientific work.

  The merits of the article: - based on broad scientific literature review - own survey research - statistical analysis of results - conclusion of survey results - the utilization of results are possible in the future work of fellow scientists, so it was worthwhile.  

Author Response

Response to Reviewer 2 comments  (attached).
